# All Screen Printed and Flexible Silicon Carbide NTC Thermistors for Temperature Sensing Applications

**DOI:** 10.3390/ma17112489

**Published:** 2024-05-22

**Authors:** Arjun Wadhwa, Jaime Benavides-Guerrero, Mathieu Gratuze, Martin Bolduc, Sylvain G. Cloutier

**Affiliations:** 1Department of Electrical Engineering, École de Technologie Supérieure, 1100 Notre Dame Street West, Montreal, QC H3C 1K3, Canada; arjun.wadhwa.1@ens.etsmtl.ca (A.W.); jaime-alberto.benavides-guerrero.1@etsmtl.net (J.B.-G.); mathieu.gratuze@etsmtl.ca (M.G.); 2Department of Mechanical Engineering, Université du Québec à Trois-Rivières, 555 Boulevard de l’Université, Drummondville, QC J2C 0R5, Canada; martin.bolduc2@uqtr.ca

**Keywords:** temperature sensing, negative temperature coefficient (NTC), printed electronics, printed temperature sensors, thermistors, silicon carbide, wide band-gap semiconductor, screen printing, silver ink

## Abstract

In this study, Silicon Carbide (SiC) nanoparticle-based serigraphic printing inks were formulated to fabricate highly sensitive and wide temperature range printed thermistors. Inter-digitated electrodes (IDEs) were screen printed onto Kapton^®^ substrate using commercially avaiable silver ink. Thermistor inks with different weight ratios of SiC nanoparticles were printed atop the IDE structures to form fully printed thermistors. The thermistors were tested over a wide temperature range form 25 °C to 170 °C, exhibiting excellent repeatability and stability over 15 h of continuous operation. Optimal device performance was achieved with 30 wt.% SiC-polyimide ink. We report highly sensitive devices with a TCR of −0.556%/°C, a thermal coefficient of 502 K (β-index) and an activation energy of 0.08 eV. Further, the thermistor demonstrates an accuracy of ±1.35 °C, which is well within the range offered by commercially available high sensitivity thermistors. SiC thermistors exhibit a small 6.5% drift due to changes in relative humidity between 10 and 90%RH and a 4.2% drift in baseline resistance after 100 cycles of aggressive bend testing at a 40° angle. The use of commercially available low-cost materials, simplicity of design and fabrication techniques coupled with the chemical inertness of the Kapton^®^ substrate and SiC nanoparticles paves the way to use all-printed SiC thermistors towards a wide range of applications where temperature monitoring is vital for optimal system performance.

## 1. Introduction

Temperature sensing is crucial in key industries such as automotive [1], health [2,3], aerospace [4], agriculture [5] and consumer electronics [6]. Temperature is a key variable in controlling and monitoring the intended function of a system [7] and must be measured to optimize process yields [8]. The three predominant types of printed temperature sensors are: 1. resistance temperature detectors (RTDs) [9], 2. thermocouples [10] and 3. thermistors [11]. Choosing the correct type of temperature sensor is dependant on the sensitivity, accuracy and temperature range required for the intended applications. Thermocouples are widely used in several industries due to their small form factor, low cost and wide temperature range. Depending on the materials of construction, some thermocouples can be used in ultra-high-temperature conditions up to 2300 °C [12]. However, when compared to thermistors, thermocouples have a lower accuracy and sensitivity, as their change in response is generally only a few milli-volts [13]. For lower temperature ranges, thermistors and RTDs are used, where RTDs typically exhibit lower sensitivity and have a slower response time compared to thermistors [13]. Thermistors on the other hand, provide high sensitivity ranging between 2 and −6%/°C [14], which makes them highly desirable for sensing across a wide variety of applications [15]. Additionally, thermistors exhibit a non-linear negative temperature coefficient (NTC) as electrical resistance decreases with increased temperature [16]. The thermal index β of a thermistor is an indicator or its sensitivity. Devices with high β values (3000–5000 K) are typically used for high-temperature sensing applications, while those with low β values (14–170 K) are used for applications such as integrated circuit temperature compensation, random access storage memories, etc. [14,17].

Several fabrication techniques, such as microfabrication [18], tape casting [19], etc., have been employed in the fabrication of thermistors. With advances in the field of printed electronics over the past two decades, it has become increasingly possible to fabricate low cost, flexible temperature sensors [20]. Literature suggests a significant increase in academic articles toward printed thermistors utilizing a variety of materials, substrates and printing methods for a different applications. Printing techniques such as screen [16], inkjet [21] and aerosol jet [22] printing have been successfully employed to fabricate all printed thermistors. Polymeric sensing materials such as poly(3,4-ethylenedioxythiophene)-poly(styrenesulfonate) (PEDOT:PSS) [23] and polydimethylsiloxane (PDMS) [24] have been widely printed to fabricate low-cost thermistors, along with carbon derivatives such as CNTs [16], graphene oxide [25] and reduced graphene oxide [26]. Semi-metallic graphene is a unique material that has been widely employed for printed thermistors [17] and RTDs [27]. The above mentioned sensing materials are generally restricted to relatively low temperature sensing ranges, generally below 100 °C, due to thermal degradation. The use of ceramics combined with higher temperature-sustaining substrates such as Kapton^®^ promises a significant increase in the temperature sensing range of the printed device.

Various ceramic materials have been employed toward the fabrication of wide-range temperature sensors, some having a sensing range as high as 1500 °C [28,29,30] using complex low and high-temperature co-firing techniques (LTCC and HTCC). Transient metal oxides such as $Ni_{x}Mn_{2-x}O_{4}$ [31], MnNiCuO [32] and $BiFeO_{3}$ [14] have been used to fabricate NTC thermistors. The fabrication of ceramic-sensing devices usual employs ultra high sintering temperatures and inert environments, which increases the overall fabrication cost and complexity [33]. Wide band-gap semiconducting materials, such as ZnO (II–VI), CaN (III–V) and SiC (IV–IV), exhibit excellent electrical and mechanical properties along with chemical inertness, and optical transparency which make them ideal candidates for flexible electronics devices [34]. Out of these, Silicon Carbide (SiC) has been gaining significant interest in recent years towards fabricating bio-sensing devices [35] owing to its bio compatibility [36,37,38]. Silicon Carbide comprises covalent bonded Si and C atoms with very short bond lengths of 1.89 Å [39], which are attributed to their mechanical and chemical stability, with an electronic band-gap ranging between 2.4 and 3.2 eV depending on the polytype [35]. Silicon Carbide exists in several polytypes, out of which cubic 3C-SiC (β-SiC), 4H-SiC and 6H-SiC (α-SiC) are most commonly grown and used for sensing applications. However, β-SiC is widely available in high purity and relatively low cost nanoparticle form. Silicon Carbide has been widely used to fabricate NTC thermistors that are predominately used in ultra-high-temperature and harsh environments. These thermistors are generally fabricated via processes such as chemical vapor depositon (CVD) [40,41], epitaxial SiC crystal growth [42], sputter-coated electrodes on SiC single crystal wafers [43] and transfer-based [44] techniques.

SiC has been sparsely used in the field of printed electronics. Researchers have demonstrated direct ink writing of SiC in borosiloxane-colloidal dispersions for microwave optics [45], inkjet printable SiC ink [46] and vat polymerization-based electrically conductive SiC features [47]. SiC nanoparticles have been employed towards applications such as electrochemical [48,49], gas [50] and humidity [51] sensing applications. There has been very limited work done towards printed SiC nanoparticles towards temperature sensing applications. In 2022, Aljasar et al. [52] demonstrated laser sintered SiC nanoparticle temperature sensors via drop casting up to 86 °C.

In this study, we focus on the fabrication of low-cost and flexible screen-printed SiC thermistors for a wide temperature range between 25 °C and + 170 °C. For this purpose, SiC nanoparticles are impregnated into a polymeric matrix of 4,$4^{\prime }$-oxydianiline (polyimide) resin via simple dispersion techniques. Initially, we study the characteristics of the commercially sourced SiC nanoparticles to assess their crystalline composition. Next, we measure electronic properties of the printed SiC sensing films and determine the optimal loading fraction of particles in the ink. The fully fabricated devices are then tested for temperature sensing performance, durability and long-term stability. The dependency of the SiC thermistors on relative humidity and mechanical deformation (bend testing) will also be investigated. Lastly, we calculate key thermistor performance matrices such as the temperature coefficient of resistance (TCR), thermal coefficient (β-index) and activation energy (eV).

## 2. Experimental Section

### 2.1. Materials

Cubic Silicon Carbide (3C-SiC, beta, 99+%, <80 nm, cubic) nanoparticles were acquired from US Research Nanomaterials, Inc., Houston, TX, USA (US2022). Poly(pyromellitic dianhydride-co-4,4-oxydianiline) amic acid resin was procured from Millipore Sigma, Burlington, MA, USA (Product: 575771); 0.005-inch thick FPC Kapton^®^ was sourced from American Durafilm, Holliston, MA, USA; 0.002-inch thick, adhesive backed Kapton^®^ was procured from Mcmaster-Carr, Elmhurst, IL, USA (Product: 2271K72). Henkel (Düsseldorf, Germany) supplied Loctite^®^ EDAG 725A silver screen printing paste. No modifications were made to any of the materials upon receipt.

### 2.2. Ink Formulation

The screen-printable thermistor ink was formulated by incorporating varying quantities of SiC nanoparticles into the poly(pyromellitic dianhydride-co-4,4-oxydianiline) amic acid solution, ranging from 30 to 40 wt.% in 5 wt.% increments. To achieve homogeneous dispersion, a two-step dispersion process was employed. The SiC nanopowder was initially weighed and added in approximately four portions to the resin. Following each addition, the ink was mixed using a planetary mixer (Thinky ARE-310, Laguna Hills, CA, USA) for three cycles of one minute each, with one minute degassing intervals at 2000 rpm. This meticulous process ensured thorough blending of the nanopowder and resin, resulting in a stable dispersion with a shelf life exceeding 6 months.

### 2.3. Device Design and Fabrication

The presented sensor structure includes two main functional layers: 1. the printed silver inter-digitated electrodes and 2. printed SiC active sensing layer. Three devices were printed for each test point to ensure repeatability. Initially, the Kapton^®^ substrate was prepared by cleaning with 99.9% pure acetone (Millipore Sigma 270725). Silver paste was printed on to the Kapton^®^ substrate using a KEKO P250 (Žužemberk, Slovenija).

Automatic screen printer with a 325 mesh, 0.001-inch emulsion thickness screen in an interdigitated electrodes (IDEs) format. The IDEs had a trace width and spacing of 0.5 mm, as depicted in Figure 1a. The silver ink was cured in air at 300 °C for 60 min in a Mancorp, Montgomery County, PA, USA (MC301N) reflow oven. Thermistor inks were then applied to the cured IDEs in 7.5 mm diameter circles and cured in the reflow oven for 60 min at 200 °C to achieve complete polymerization of the resin. Under these conditions, a polycondensation reaction occurs between a diamine and a dianhydride, resulting in the formation of polyimide [14]. Lastly, a 0.002-inch thick adhesive backed polyimide film was applied atop two of the printed sensors to inhibit the influence of atmospheric humidity. Figure 2a–c illustrate the fabrication process of the SiC thermistors. High-resolution optical microscpe images of the printed SiC thermsitor can be seen in Figure 2d.

### 2.4. Characterization Methods

X-ray diffraction patterns were obtained using the XRD-$X^{\prime }Pert^{3}$ system with a cobalt source, and Raman micro-spectra of the SiC nanopower were obtained using the WITec alpha300A (Ulm, Germany) raman microspectoscopy system with a 532 nm green laser. Ultraviolet-visible (UV-Vis) absorption spectra of the SiC nanopowder was obtained from via the Perkin Elmer (Waltham, MA, USA), Lambda 750 system. Micrographs of the printed films were obtained using a Hitachi SU8230 (Tokyo, Japan) scanning electron microscope (SEM) equipped with a Bruker (Billerica, MA, USA), QUANTAX FlatQUAD EDX detector for precise elemental mapping. Transmission electron microscopy on the nanoparticles was performed via the Jeol (Akishima, Japan), JEM-F200 multi-purpose electron microscope. A Keyence VHX7000 (Osaka, Japan) optical microscope was used to obtain high-resolution images of the sensor. Ossilla T2001A3 (Sheffield, UK) four-point-probe was utilized to measure the conductivity of the printed SiC films. The printed thermistors were tested in a controlled temperature and humidity 6-channel micro probe station from Nextron^®^ (MPS-PT6C) (Tokyo, Japan), and the two-wire resistance readout was recorded using a 40-channel digital multi-meter from Keithley, Cleveland, OH, USA (DMM 2790-L). Temperature varied between 25 °C and 170 °C owning to the maximum temperature limit of the Nextron^®^ at a constant rate of 10 °C/min and a constant humidity at 40% RH. A custom Matlab script was written to process the raw data obtained from both the humidity test chamber and digital multi-meter. Figure 3 illustrates the experimetal setup used. Current–voltage characteristics of the thermistors were obtained using the Keithley DAQ6510 digital multi meter and the Keithley 2400 source-measure unit. Mechanical bend testing was performed using a custom setup consisting of a liner stage attached to a ball screw stepper motor. The stage was controlled via a Arduino Uno and a custom script to control travel distance, speed and number of cycles.

## 3. Results and Discussion

### 3.1. Material Characterization

The X-ray diffraction (XRD) patterns of the commercially purchased SiC particles validate their cubic crystalline nature, exhibiting well-defined and sharp diffraction peaks, as depicted in Figure 4a. The peaks observed at 2θ = 39.6°, 42° and 71.2° correspond to the 002, 200 and 311 crystalline planes of the 3C−SiC cubic crystal structure, consistent with the (International Centre for Diffraction Data) ICDD card: 04-002-9070 [53,54]. Additionally, the presence of 2H-SiC peaks at 2θ = 45° (015) and 49° (016), as per the ICDD card: 04-008-2392 [55], may potentially be attributed to a blending of the two phases formed during the nanoparticle fabrication process [56]. The spectral results suggest an approximate 9:1 ratio between the two phases, respectively.

The Raman spectra of the 3C−SiC powder (Figure 4b) confirm its cubic crystal structure. Noteworthy peaks appear at wave numbers 785, 897, 1346, 1588, 2684 and 2916 ${\mathrm{cm}}^{-1}$. Specifically, the peaks at 785 and 897 ${\mathrm{cm}}^{-1}$ correspond to the transverse and longitudinal optical modes of 3C−SiC, as documented in reference [57,58]. The 1346 ${\mathrm{cm}}^{-1}$ peak represents the D band of carbon, peak 1588 ${\mathrm{cm}}^{-1}$ corresponds to the G band, associated with the $A_{1g}$ vibrational mode of carbon. The peak at 2684 ${\mathrm{cm}}^{-1}$ signifies the second orders of the D band, referred to as the 2D band, while the 2916 ${\mathrm{cm}}^{-1}$ peak represents the D + G band, as detailed in references [59,60].

Ultraviolet–visible–near infrared (UV-vis-NIR) absorption spectroscopy was performed on the 3C−SiC powder (Figure 4c), which indicated that the absorption edge is close to 400 nm. The electronic band-gap ($E_{g}$) of the nanopowder was calculated by linear fitting of the Tauc plot, as seen in the inset graph. It suggests the band gap of the 3C−SiC nanopowder is 2.75 eV, which is consistent with previous reports [61,62,63].

### 3.2. Morphology

The morphology of the 3C−SiC particles is investigated via scanning electron microscopy (SEM) micrographs presented in Figure 4d. We observe a densely packed film of particles with minimal porosity. The image processing software ImageJ was used to estimate the film porosity to be approximately 9.74%. Elemental mapping of the particles (Figure 5) revealed a uniform distribution of silicon and carbon elements along with the presence of elemental oxygen, which could be attributed to the native oxide shell around the SiC particles. Table A1 in Appendix A provides the mass fractions of each element observed.

Transmission electron microscopy micrographs are shown in Figure 6a,b. The SiC particles have an average particle size of 70 μm, as analyzed using ImageJ. We confirm the presence of an approximately 2 nm thick oxide shell (SiO_x_) around the SiC particles, as indicated in Figure 6b [64]. The SAED pattern observed in Figure 6c allows us to calculate the inter planer atomic distance to be 0.25 nm, which is assigned to the (111) cubic atomic plain [65].

### 3.3. Printed SiC Thermistor Characterization

Once the 3C−SiC particles have been thoroughly characterized, the screen-printable thermistor ink is fabricated as described in Section 2.2 and printed directly atop the prepared Kapton^®^ substrate to determine the electrical properties. Figure 7a shows the current voltage characteristics of the thermistor at 25 °C. We observed a linear ohmic behavior between − 5 V and 5 V, indicating that the device functions as a resistor where the current is directly proportional to the applied voltage. Further, we observed a negligible contact resistance of 0.0028 Ω at the Ag-SiC interface, which was determined as shown in Figure A2 in Appendix A.

Next, we measured the electrical conductivity of the three inks via four point probe. As seen in Figure 7b, the 30 wt.% loaded SiC thermistor ink formulated exhibits a maximum electrical conductivity of 2.99 ± 0.007 S/m. We observe that electrical conductivity decreases by more than 50% between the 30 wt.% and 35 wt.% loaded samples and further reduces as the loading is increased to 40 wt.%. This reduction could be attributed to the onset and growth of micro-cracks on the surface of the SiC film during the annealing process [66]. This phenomena was further investigated by acquiring SEM micrographs of films printed with the three SiC loaded inks at two different magnifications. The 30 wt.% SiC film has a dense, self leveled and uniform distribution of particles with no cracks on the surface of the film (Figure 8a,b). As the SiC loading is increased to 35%, we observe the formation of surface imperfections and agglomerates, as seen in Figure 8c, and the onset of micro cracks in the films, which are observed at high magnifications (Figure 8d). Lastly, the cracks are significantly more pronounced in the 40 wt.% loaded SiC film (Figure 8e,f). To further optimize the wt.% loading fraction, we studied the evolution of micro-cracks on the surface of 31, 32, 33 and 34 wt.% loaded SiC films, as seem in Figure A1 in Appendix A. Indeed, we see a progressive increase in the quantity and size of cracks as the loading fraction is increased. Hence, the decrease in electrical conductivity with increase in wt.% SiC loading can directly be attributed to film cracking due to the loss of conductive pathways [67,68].

The silver IDEs were screen printed onto the polyimide substrate as previously described, and the SiC inks were then screen printed atop the silver IDEs to fabricate the thermistors, as described in Section 2.3 and Figure 2.

Although the Kapton^®^ substrate is rated to handle 350 °C and up to 400 °C for intermittent exposure [69], the performance of the printed SiC thermistor is restricted to 170 °C due to the Nextron micro-probe stations’ maximum temperature limit. The thermistors of each ink formulation were thermally cycled between 25 °C and 170 °C five times, and the electrical resistance was recorded. Figure 9a shows the change in electrical resistance of all three inks with increases in temperature over five test cycles. Amongst the three ink formulations, the 30 wt.% loaded SiC thermistor showed the highest change in resistance of 41.1% going from 1267 Ω at 25 °C 988 Ω at 170 °C. The 35 wt.% and 40 wt.% SiC showed a considerably low change of 20.4% and 14.7%, which is less than half of the 30 wt.% ink. Interestingly, the response of the three inks is identical to its change in electrical conductivity, as previously discussed in Figure 7b. This further validated that the increase in SiC loading significantly contributes to the degradation in the thermisors performance due to film cracking. The best performing thermistor (30 wt.% SiC) follows a second-order polynomial fitting governed by Equation (Equation 1) with an $R^{2}$ confidence of 0.96.
(1)$$y=-2\times 10^{-0.5}X^{2}+0.0064X+1.0714$$

Similar to conventional SiC thermistors reported in the literature [40], our screen printed SiC thermistors show a reduction in electrical resistance with increases in temperature, exhibiting a negative temperature coefficient of resistance (NTC). Figure A3 in Appendix A shows the response of the three individual 30 wt.% thermistors cycled five times each. We observed excellent repeatability across different sensors and multiple cycles.

A high-performance thermistor must be stable, reliable and durable to be successfully deployed in challenging application environments [13]. To this effect, we subject the 30 wt.% SiC thermistor to various endurance tests. In Figure 9b, a change in resistance of the thermsistor was recorded over 15 h while being cycled between 25 °C and 170 °C. The device experiences minor variability during the first two thermal cycles, which could be attributed to the relaxation of the SiC film and any instabilities in the temperature test chamber at the onset of the test. Beyond this, the device is extremely stable with a minor drift in baseline resistance, which was corrected via background subtraction. In order to test device stability, the 30 wt.% SiC thermistor is maintained at 25 °C, 50 °C, 100 °C, 150 °C and 170 °C for 1 h each. We observe constant resistance at each fixed temperature, suggesting device stability over a prolonged time duration (Figure 9c).

The influence of humidity has a significant impact on the performance of a thermistor [70]. Researchers have shown that SiC exhibits a good humidity response, which is well suited to making humidity sensors [71,72]. To mitigate this effect, we have incorporated a polyimide film encapsulation layer. To ensure its effectiveness, we measure the electrical resistance of the 30 wt% SiC thermistor while varying the relative humidity between 10%RH and 90%RH at 25 °C. The change in resistance of two identical devices—one with and one without the polyimide film encapsulation—are compared as seen in Figure 9d. The device without encapsulation experiences a 32.6% change in resistance at a constant temperature, indicating that the SiC film is highly sensitive to changes in relative humidity. In contrast, the device with encapsulation sees a much smaller drift of 6.5% in baseline resistance under the same conditions of %RH. The laminated polyimide encapsulant provides a high degree of protection against %RH; however, this could be further improved by incorporating a fully printed insulating layer in future studies.

The mechanical flexibility of the 30 wt.% SiC thermistors was tested at an aggressive angle of 40°, as seen in the inset of Figure 9e. We observed a small drift in baseline resistance of 4.2% over 100 test cycles (Figure 9e). Interestingly, the majority of the change in resistance was observed during the first 20 cycles, indicating deformation in the SiC sensing film at the onset of the test. Post bending, the device was cycled five times between 25 °C and 170 °C and compared to a pristine 30 wt.% device (Figure 9f). We notice a reduction in device response from 41.1% to 26.2%. This drop in performance is tentatively attributed to surface crack formation in the SiC film, leading to loss of conductive pathways for effective change transfer with changes in temperature.

### 3.4. Thermistor Performance

To further characterize the 30 wt.% SiC thermistor, we calculate its performance parameters such as thermal index β, activation energy $E_{g}$ and temperature coefficient of resistance (TCR). The electrical resistance dependence on temperature is given by the expression $R=R_{o}exp\left(\beta /T\right)$, where $R_{o}$ is the resistance at infinite temperature and *T* is the absolute temperature. Thus, the thermal index can be calculated using the following [73].
(2)$$\beta =\frac{lnR_{1}-lnR_{2}}{\left(1/T_{1}\right)-\left(1/T_{2}\right)}$$Here, $T_{1}$ = 248 K, $T_{2}$ = 438 K and $R_{1}$ and $R_{2}$ are the resistance values in Ω at the respective temperatures. The activation energy is calculated from the expression $E_{g}$ = 2kβ, where k is the Boltzmann constant, and the TCR is calculated by the expression [14].
(3)$$TCR=-\frac{\beta }{T^{2}}$$

Using Equations (Equation 2) and (Equation 3), the characteristics of the printed thermistors are calculated as presented in Table 1. For the 30 wt.% SiC thermistor, the thermal index observed is 502 ± 11 K, the TCR value of −0.556 ± 0.012%/°C with an activation energy of 0.08 ± 0.001 eV.

The accuracy of the printed 30 wt.% SiC thermistor was determined using the Steinhart’s equation (Equation (Equation 4)) [16,74].
(4)$$\frac{1}{T}=A+BlnR+C{\left(lnR\right)}^{3}$$
where *T* is the temperature in Kelvin, *R* is the resistance in Ω at temperature *T*. *A*, *B* and *C* are the Steinhart coefficients, which are material specific and are used to calibrate the SiC thermistor accuracy. To this effect, experimentally measured resistance values at three temperatures of 25 °C ($T_{1}$ = 298 K), 100 °C ($T_{2}$ = 373 K) and 170 °C ($T_{3}$ = 443 K) are used in Equation (Equation 5).
(5)$$\left[\begin{array}{ccc} 1 & ln(R_{1}) & ln^{3}\left(R_{1}\right)\\ 1 & ln(R_{2}) & ln^{3}\left(R_{2}\right)\\ 1 & ln(R_{3}) & ln^{3}\left(R_{3}\right) \end{array}\right]\left[\begin{array}{c} A\\ B\\ C \end{array}\right]=\left[\begin{array}{c} \frac{1}{T_{1}}\\ \frac{1}{T_{2}}\\ \frac{1}{T_{3}} \end{array}\right]$$

For the 30 wt.% SiC thermistor, the Steinhart’s coefficients are *A* = −8.12 × $10^{-2}$, *B* = 1.59 × $10^{-2}$ and *C* = −8.12 × $10^{-5}$, with an accuracy of ± 1.35 °C over the entire tested temperature range (25 °C to 170 °C) (Figure 10). The accuracy of the fully printed SiC thermsitors is comparable to conventional SiC and commercially available highly accurate thermistors ranging between ± 0.05 °C to ± 1.5 °C [75,76,77].

Researchers have developed several SiC-based thermistors in recent years that have superior performance compared to our all-printed SiC nanoparticle-based thermistors (highlighted in Table 2). However, these thermistors are fabricated via complex and expensive processes such as chemical vapor deposition, vacuum vapor deposition and sputtering. The overall cost of fabrication and material use limit the use of such thermistors to only highly specialized applications. On the contrary, the SiC thermistors developed in this study employ screen printing, which is an inexpensive and scalable technology. Additionally, the use of commercially sourced SiC particles, polyimide resin and Kapton^®^ substrate makes these devices relatively extremely low cost to fabricate. When compared to other printed thermistors recently reported in the literature (Table 2), the 30 % SiC thermistors exhibit a comparable TCR coefficient over such a wide operational temperature range. Both SiC particles and Kapton^®^ are chemically inert [78], bio-compatible [79] and resistant to ultra-high temperatures [80]. These properties, combined with the simplicity and flexibility of our SiC thermistors, allow them to accurately detect temperature variations in critical applications such as medicine, agriculture, aerospace and chemical production.

## 4. Conclusions

In this study, we showcase a high performance, fully printed and flexible silicon carbide-based thermistor for wide temperature range applications. Low-cost serigraphic screen printing was used to fabricate sensors onto silver-printed IDEs on flexible Kapton^®^ substrate.

SiC thermistors were cycled over a wide temperature sensing range between 25 °C and 170 °C. Three inks with different SiC ink loading’s were tested and optimal device performance was achieved at 30 wt.% loading.At higher loading of 40 wt.%, we observed a reduction in device performance, which was attributed to this onset and propagation of cracks within the printed film, leading to loss of conductive pathways.The 30 wt.% device exhibits a TCR of −0.556%/°C, along with a thermal index of 502 K (β-index) and an activation energy of 0.08 eV. The devices exhibit excellent repeatability and reliability after cycling over extended periods of time up to 15 h.Printed thermistors shows a small variation in baseline resistance of 6.5% while being tested over a wide relative humidity range (10–90%RH).Aggressive bend testing was performed to test the thermistor flexibility. We observed a small 4.2% drift in baseline resistance of the device after 100 bend cycles at 40 °C.Lastly, the 30 wt.% printed thermistors exhibit an accuracy of ± 1.35 °C, which is at par with commercially available high-accuracy thermistors.

This study demonstrates that flexible, all screen printed SiC thermistors have an immense potential in the field of flexible and printed electronics. These low cost, flexible and mass producible thermistors can help improve critical user–device interactions in urgent application such as healthcare, agriculture and food monitoring.

## Figures and Tables

**Figure 1 materials-17-02489-f001:**
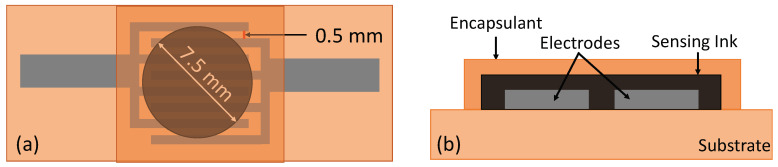
SiC thermistor schematic: (**a**) top view, (**b**) side view.

**Figure 2 materials-17-02489-f002:**
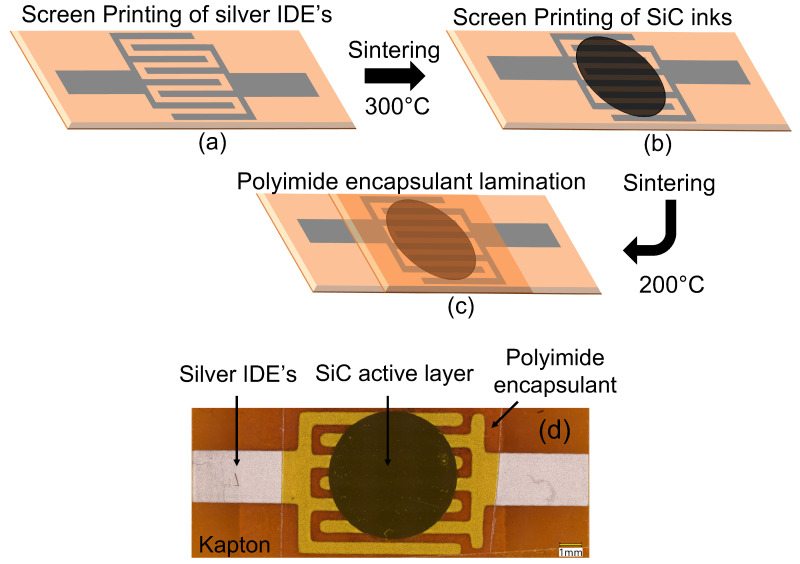
(**a**–**c**) SiC thermistor fabrication process. (**d**) High-resolution optical microscopy image of printed SiC thersmitor.

**Figure 3 materials-17-02489-f003:**
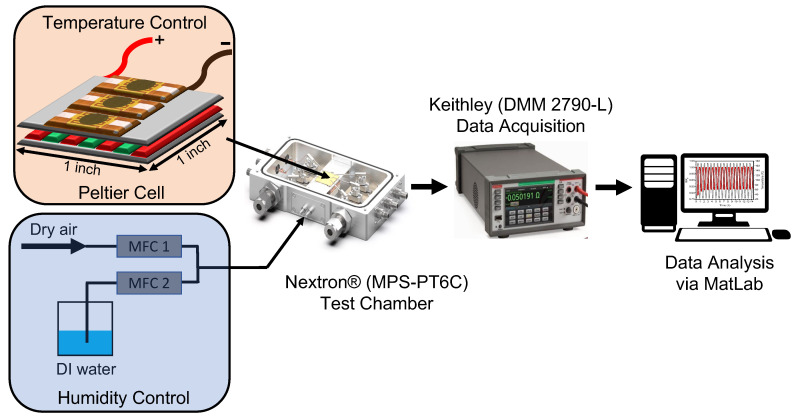
Experimental setup.

**Figure 4 materials-17-02489-f004:**
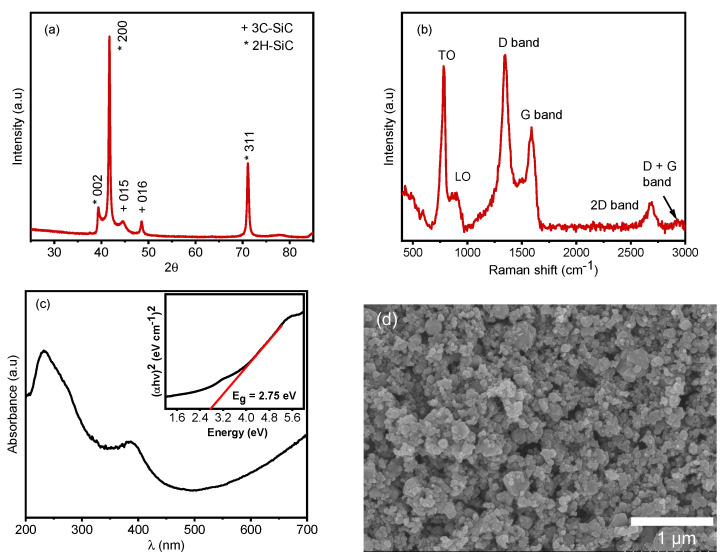
Characterization of commercially sourced 3C−SiC particles. (**a**) XRD spectra, (**b**) Raman spectra, (**c**) UV-VIS spectra with Tauc plot as inset to determine electronic band gap, (**d**) SEM micrograph.

**Figure 5 materials-17-02489-f005:**
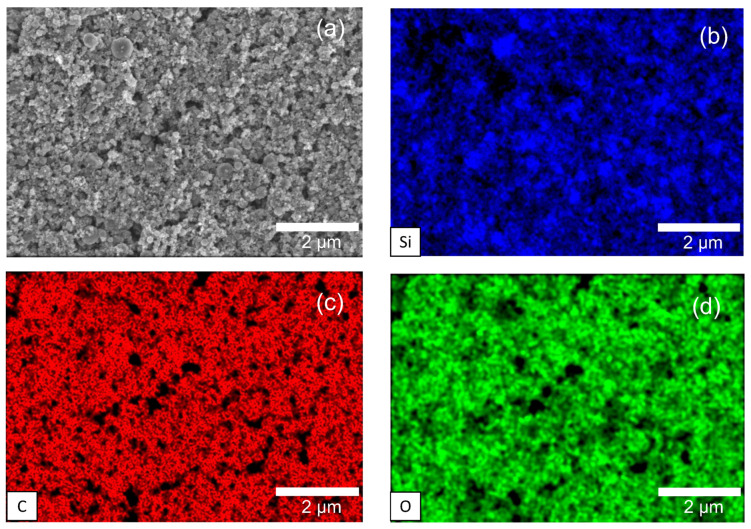
EDX imaging of (**a**) SiC particles representing: (**b**) Si, (**c**) C and (**d**) O species.

**Figure 6 materials-17-02489-f006:**
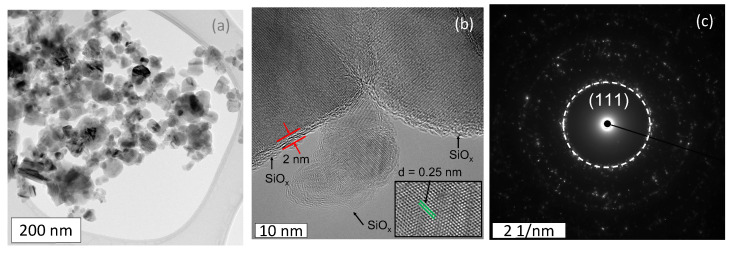
(**a**,**b**) TEM micrographs and (**c**) SAED pattern of SiC particles.

**Figure 7 materials-17-02489-f007:**
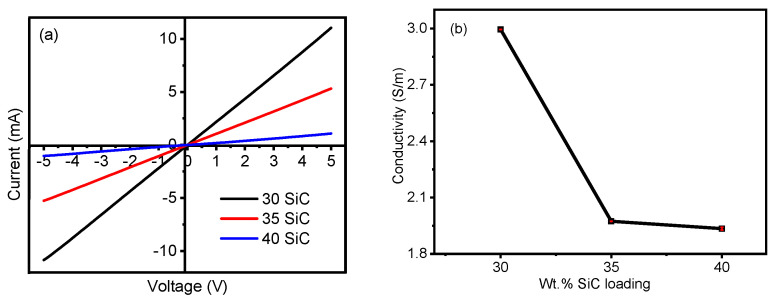
(**a**) Current–voltage characteristics and (**b**) electrical conductivity of 30 wt.%, 35 wt.% and 40 wt.% SiC thermistor inks.

**Figure 8 materials-17-02489-f008:**
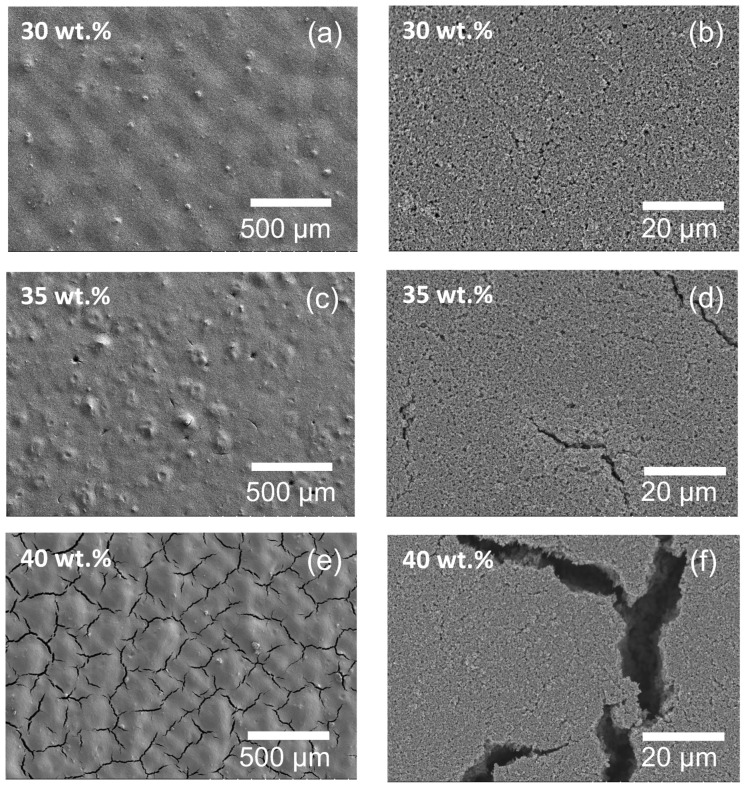
SEM micrographs of (**a**,**b**) 30 wt.%, (**c**,**d**) 35 wt.% and (**e**,**f**) 40 wt.% SiC thermistor inks.

**Figure 9 materials-17-02489-f009:**
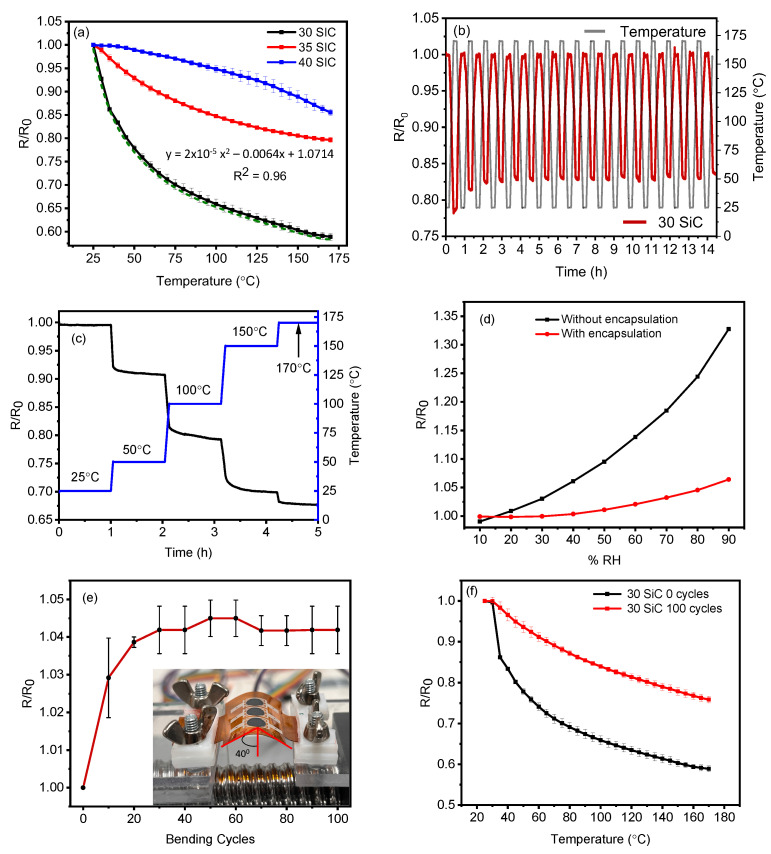
(**a**) Electrical resistance versus temperature of 30 wt.%, 35 wt.% and 40 wt.% SiC thermistors. (**b**) Long-term cycling, (**c**) thermal stability, (**d**) humidity stability, (**e**) change in baseline resistance after bend testing and (**f**) electrical resistance versus temperature plot after bend testing of 30 wt.% SiC thermistor.

**Figure 10 materials-17-02489-f010:**
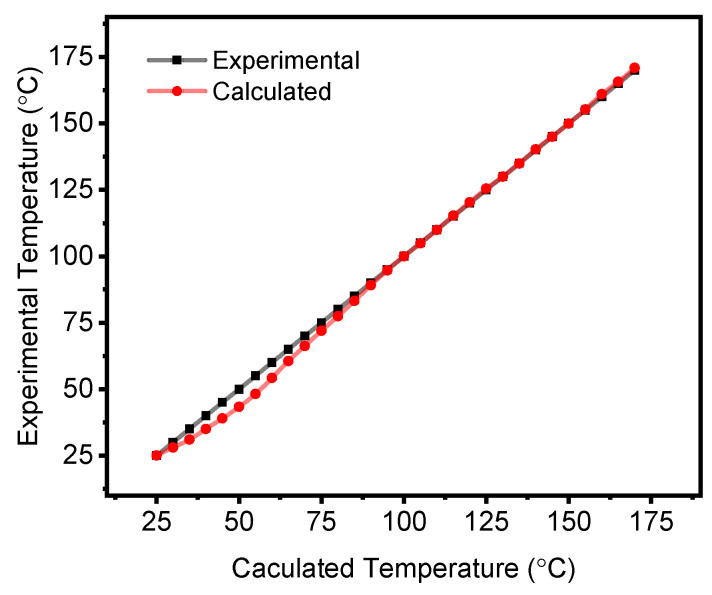
Accuracy of printed 30 wt.% SiC thermistor calculated via the Steinhart’s equation.

**Table 1 materials-17-02489-t001:** Electrical performance of various printed SiC thermistor inks.

Ink Name	3C−SiC wt%	PI Resin wt%	TCR (%/°C)	Activation Energy (eV)	Thermal Index (K)
30-SiC	**30**	**70**	**−0.556 ± 0.012**	**0.08 ± 0.001**	**502 ± 11**
35-SiC	35	65	−0.227 ± 0.008	0.035 ± 0.001	205 ± 7.4
40-SiC	40	60	−0.157 ± 0.0068	0.025 ± 0.001	142 ± 6

**Table 2 materials-17-02489-t002:** State-of-the-art of SiC-based and printed thermistors.

Sensing Material	Electrodes	Fabrication Method	Temperature Range	TCR	Reference
Silicon Carbide based thermistors					
On SiC undoped wafer	Al, Au, Pt-Pd alloy	CVD	25 °C–400 °C	−7.9%/K	[40]
Anodic bonding SiC on glass	Ni/Al	CVD	300 K–600 K	−1.3%/K at 300 K −0.3%/K at 600 K	[44]
n-type single crystal 21R-SiC wafer	Au-$TiB_{x}$-Ni	magnetron sputtering	77 K–450 K	$10^{5}$ times change in resistance	[43]
Doped unicrystalline SiC wafer	W (Tungsten)	welding	−100 °C–300 °C	1.9%/°C	[62]
Polycrystalline CVD-SiC wafer	Pt-Pd alloy	Vacuum vapor deposition	25 °C– 365 °C	−5.5%/°C	[81]
Printed Thermistors					
Carbon nanotubes	Ag	Screen printing	−40 °C–100 °C	−0.4%/°C	[16]
PDMS + graphene	Copper wire	3D printing	25 °C–70 °C	0.008%/°C	[82]
graphene oxide	Ag	Inkjet printing	25 °C–85 °C	−1.19%/°C	[83]
$BiFeO_{3}$ + 3.5 wt% graphene	Ag	Screen printing	25 °C–170 °C	−0.961%/°C	[14]
$V_{2}O_{5}$	Pt	Screen printing	200 K–400 K	−3.7 to −1.7%/K	[84]
3C−SiC	Ag	Screen printing	25 °C–170 °C	−0.556%/°C	**This work**

## Data Availability

The original contributions presented in the study are included in the article, further inquiries can be directed to the corresponding author.

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
