# Peer review of "All Screen Printed and Flexible Silicon Carbide NTC Thermistors for Temperature Sensing Applications"

_materials, 2024, doi:10.3390/ma17112489_

Round 1
Reviewer 1 Report
Comments and Suggestions for Authors
The authors developed screen printed and flexible silicon carbide NTC thermistors for temperature sensing applications. The work is interesting but should be revised before publication.
1. The Introduction should be shortened and highlight the novelty of this work. The presentations are partly repetitive with those of Conclusion.
2. Please check the coordinates of the graph of Fig. 10.
3. The merit of this method should be compared with that of others reported previously.
4. The English should carefully checked, For example, “and monitoring the intended function of a system [7] to and to optimize process yields” “Carbon derivaties such as carbon nanotubes (CNT’s) [16], graphene oxide [25] and reduced graphene oxide [26]. have also been utilized for the same.” “…commercial silver ink was screen printed to fabricate…”.
Comments on the Quality of English LanguageMinor editing of English language required.
Reviewer 2 Report
Comments and Suggestions for Authors
The current manuscript studies the characteristics of screen printed and flexible silicon carbide thermistors which are related to temperature sensing applications. The application is important and the manuscript is well organized and written. However, there are some issues to be considered as follows:
- The abstract should be focused and reduced to be within 150-200 words.
- It is important to illustrate how many samples were fabricated and tested, and what about the standard deviation obtained from the measurement tests.
- The authors stated that the current method is a cost-effective compared to other methods. Is it efficient and compatible with the industrial production or was it applied in order to fabricate the research samples?
- The conclusion should include the main results, contributions, and novelty of the current work, in addition to the future perspective. Using a pullet points style is highly recommended.
Comments on the Quality of English LanguageModerate edit is required.
Reviewer 3 Report
Comments and Suggestions for Authors The results are good, and the manuscript is well written. The work and presentation are commendable. I have a few minor comments. Line 57 may be a comma instead of a full-stop The scale bar is a bit confusing in Figure 2d Possibly use a scale bar in Figure 3 with either the device or the 6-probe setup. Can you quantify the porosity? line 191 Also, quantify the individual elements along with the color maps (Figure 5) Do you think the crack was observed at any wt% between 30 and 35? like 32, 33? It may be interesting as an extension to see how to engineer the film better. Can you comment if ts process can be used on other substrates also ? like some other temperature resistant tapes/films?Author Response
please see attached

Reviewer 4 Report
Comments and Suggestions for Authors
The paper reports on the use of SiC nano-particles for NTC temperature sensing.
The paper is well structured. There is sufficient background on previous published work. The methods are clearly described. There is useful analysis of the material using variety of techniques. The material recipe has been optimized. The device has been well characterized, including over temperature, humidity and time. The results presented appear to support the conclusions.
Could the authors comment on electrode contact resistance and whether this is an issues with?
Some comment on the underlying principle and physics of operation would be useful.
Fig 7a: It would be useful to have the measurement points marked on the plot.
Fig 9 a): Could the authors comment on the differences in the resistance – temperature profiles?
Table 2: Some resistance-temperature coefficients are given in ppm, others %/K. Perhaps the units could be unified.
Comments on the Quality of English LanguageThe quality of English is high.
Round 2
Reviewer 1 Report
Comments and Suggestions for Authors
Accept
Reviewer 2 Report
Comments and Suggestions for Authors
The revised manuscript is significantly improved. The review comments and recommendations are well addressed.
Comments on the Quality of English LanguageMinor editing is required.